# Actionable Predictive Factors of Homelessness in a Psychiatric Population: Results from the REHABase Cohort Using a Machine Learning Approach

**DOI:** 10.3390/ijerph191912268

**Published:** 2022-09-27

**Authors:** Guillaume Lio, Malek Ghazzai, Frédéric Haesebaert, Julien Dubreucq, Hélène Verdoux, Clélia Quiles, Nemat Jaafari, Isabelle Chéreau-Boudet, Emilie Legros-Lafarge, Nathalie Guillard-Bouhet, Catherine Massoubre, Benjamin Gouache, Julien Plasse, Guillaume Barbalat, Nicolas Franck, Caroline Demily

**Affiliations:** 1Centre d’Excellence Autisme iMIND, pôle HU-ADIS, Hôpital le Vinatier, 69678 Bron, France; 2Equipe «Disorders of the Brain», Institut Marc Jeannerod, UMR 5229, CNRS & Université Lyon 1, 69100 Villeurbanne, France; 3Pôle Centre Rive Gauche, Hôpital Le Vinatier, 69678 Bron, France; 4Centre Hospitalier Universitaire de Saint-Etienne, 42270 Saint-Priest-en-Jarez, France; 5Hôpital Charles Perrens, Université de Bordeaux, 33405 Talence, France; 6CREATIV & URC Pierre Deniker, Centre Hospitalier Laborit, Université de Poitiers, 86000 Poitiers, France; 7Centre Référent Conjoint de Réhabilitation (CRCR), Centre Hospitalier Universitaire de Clermont-Ferrand, 63000 Clermont-Ferrand, France; 8Centre Référent de Réhabilitation Psychosociale de Limoges (C2RL), 87000 Limoges, France; 9Centre Hospitalier Laborit, 86000 Poitiers, France; 10Faculté de Médecine, Université de Saint-Etienne, 42023 Saint-Etienne, France; 11Centre Hospitalier Alpes-Isère, 38120 Saint Egrève, France; 12Centre Ressource de Réhabilitation Psychosociale et de Remédiation Cognitive (CRR), CH le Vinatier et Institut Marc Jeannerod, UMR 5229 & Université Lyon 1, 69100 Bron, France

**Keywords:** homelessness, antipsychotics, REHABase, psychotropic medication, classification and regression tree model (CART), machine learning, depression

## Abstract

Background: There is a lack of knowledge regarding the actionable key predictive factors of homelessness in psychiatric populations. Therefore, we used a machine learning model to explore the REHABase database (for rehabilitation database—*n* = 3416), which is a cohort of users referred to French psychosocial rehabilitation centers in France. Methods: First, we analyzed whether the different risk factors previously associated with homelessness in mental health were also significant risk factors in the REHABase. In the second step, we used unbiased classification and regression trees to determine the key predictors of homelessness. Post hoc analyses were performed to examine the importance of the predictors and to explore the impact of cognitive factors among the participants. Results: First, risk factors that were previously found to be associated with homelessness were also significant risk factors in the REHABase. Among all the variables studied with a machine learning approach, the most robust variable in terms of predictive value was the nature of the psychotropic medication (sex/sex relative mean predictor importance: 22.8, σ = 3.4). Post hoc analyses revealed that first-generation antipsychotics (15.61%; *p* < 0.05 FDR corrected), loxapine (16.57%; *p* < 0.05 FWER corrected) and hypnotics (17.56%; *p* < 0.05 FWER corrected) were significantly associated with homelessness. Antidepressant medication was associated with a protective effect against housing deprivation (9.21%; *p* < 0.05 FWER corrected). Conclusions: Psychotropic medication was found to be an important predictor of homelessness in our REHABase cohort, particularly loxapine and hypnotics. On the other hand, the putative protective effect of antidepressants confirms the need for systematic screening of depression and anxiety in the homeless population.

## 1. Introduction

Homelessness is a public health issue with a broad definition that includes two subcategories: (1) individuals with no residence (living on the street or in a homeless shelter) and (2) individuals who are in transit or in temporary arrangements (homeless people living in hotels, temporary accommodations or insecure accommodations, such as living temporarily with family or friends because they have nowhere else to go) [1]. In Europe, it is estimated that as many as 4.1 million people experience an episode of homelessness each year [2]. Moreover, the prevalence of homelessness seems to have increased in recent years [3], even before the SARS-CoV-2 pandemic. According to the European Commission, homelessness is one of the most extreme and traumatic forms of social isolation [4] and is associated with a high level of perceived discrimination, victimization, and aggression—elements that are well documented as primary risk factors for mental illness [5]. The last *Lancet* Commission report on global mental health included mention of homelessness as both a cause and consequence of poor mental health.

Exploratory studies have already identified some individual risk factors for homelessness, such as male sex, non-heterosexual identity, low education levels, unemployment, being single, adverse life events in childhood and adulthood, criminal behavior, a history of running away, and a history of frequent of moves [6]. Indeed, various data indicate a high prevalence of psychiatric disorders among homeless people—up to 48.4%, according to various studies [7]. A recent meta-analysis suggests that the burden of psychiatric morbidity in homeless persons is substantial and should lead to regular reviews of how health care services assess, treat, and follow-up homeless people [8]. This evidence confirms a reciprocal and closed relationship between homelessness and mental health. However, we lack knowledge about the actionable key predictive factors leading to homelessness among individuals with psychiatric or neurodevelopmental disorders. This is a major issue, as these individuals are undoubtedly one of the populations most at risk of experiencing such difficulties.

Our strategy for identifying these critical actionable factors was as follows. First, to identify actionable factors that were easily identified in the target population. Then, to identify multi-diagnostic databases listing these factors. Finally, the core of this study was to design an analysis strategy based on machine learning and robust to the heterogeneity of the data in order to comparatively evaluate the different factors identified. The final objective was to rank the importance of the different factors identified in order to analyze in detail the essential factors to be taken into account when designing the most effective public health strategies. Indeed, this study aimed to identify the most critical predictors of homelessness to provide guidance for further research and tailored prevention and intervention programs related to mental health.

The present study was carried out using data from the REHABase, which is a cohort of patients with psychiatric or neurodevelopmental disorders referred to French psychosocial rehabilitation centers [9]. The REHABase provides a large database with a large number of diverse and heterogeneous variables; therefore, we used specific machine learning and variable selection procedures to obtain accurate and robust predictions based on the fast and efficient model of classification trees.

More specifically, to validate the REHABase data for machine learning exploration, we started by analyzing the individual risk factors commonly associated with homelessness and the various prevalence of housing deprivation across the different diagnostic categories. Indeed, some diagnoses, such as schizophrenia spectrum disorders, have well been identified as risk factors for homelessness, but knowledge about other diagnostic categories remains scarce [10,11]. In a second step, we identified which of the potential predictors recorded in the REHABase would be the most effective in identifying a substantial risk of housing deprivation using the CART classification and regression tree model [12], which is capable of managing the complexity and heterogeneity of the variables and their interactions. Finally, in line with previous literature [13], we independently explored the cognitive impairment associated with homelessness, taking into account that the REHABase data offer the advantage of a broad cognitive assessment of all users and that these additional results can contribute to a better understanding of the mechanisms at work in a psychiatric population that could lead to homelessness.

## 2. Materials and Methods

### 2.1. Database Evaluation

#### 2.1.1. Population

The REHABase cohort [9] was recruited in 2016 to collect information on users with serious mental illness or neurodevelopmental disorders attending the centers of the French psychosocial rehabilitation network. Clinically stabilized users are referred to these centers by public mental health services, private psychiatrists, or private practitioners or are self-referred. A functional and cognitive standardized evaluation is performed to establish a personalized rehabilitation care plan in collaboration with the patient. Users attending the centers are included in the REHABase if their Global Assessment of Functioning (GAF) [14] score is <61, according to the cut-off for social recovery identified by a previous meta-analysis [15]. The study was authorized by the French legislation (French National Advisory Committee for the Treatment of Information in Health Research, 16.060bis), including information processing (French National Computing and Freedom Committee, DR-2017-268).

#### 2.1.2. Baseline Assessment

In all centers of the network, the baseline assessment is routinely performed by a multidisciplinary team specialized in psychosocial rehabilitation, including psychiatrists, nurses, neuropsychologists, and occupational therapists. Demographic, clinical, functioning, and cognitive data were collected by using a standardized electronic case report form. Regular group meetings are held to select the instruments used for the clinical and cognitive evaluations, monitor quality control, and ensure suitable interrater reliability [9].

#### 2.1.3. Clinical and Functioning Measures

Before inclusion in systematic REHABase data collection, users are first given a diagnosis based on the specific clinical interview (MINI) from DSM-5 [16]. Those with autism spectrum disorder (ASD) were diagnosed using the Adult Asperger Assessment [17] or Autism Diagnostic Interview [18] and Autism Diagnosis and Observation Schedule [19].

The scales used in the REHABase were chosen by consensus among the first centers that participated in the project. We used data from the following scales: (i) the GAF scale [14]: a clinician-rated global measure of psychological, social, and occupational functioning (score 1–100, high score indicates better functioning); (ii) the Clinical Global Impression severity (CGI-S) scale [20]: a clinician-rated measure of the severity of illness (score 1–7; high score indicates greater severity); (iii) medical treatment compliance was assessed using the Medication Adherence Rating Scale (MARS) [21], a ten-item self-report questionnaire; and (iv) insight was investigated by the Birchwood Insight Scale (BIS) [22], a self-reported self-perception of mental illness questionnaire.

A total of 3416 users who were effectively evaluated at baseline at the time of extraction in May 2021 were included in the study. Considering the main diagnosis, most of the users met the criteria of schizophrenia spectrum disorder (45.70%); 12.67% of them had a neurodevelopmental disorder (81.49% among them had ASD) (Table 1). A total of 63.88% of the participants were male, and the mean age was 33.03 years (SD = 10.71).

### 2.2. Homelessness in the REHABase Database

Our primary variable of interest was homelessness, a binary variable in our database that can refer to people (1) sleeping on the streets and (2) residing in shelters, temporary housing, or precarious housing. This definition was edited by a group of experts on Population and Housing censuses in 2009 at the United Nations Economic Commission for Europe Conference of European Statisticians. Among the 3416 users in the database, 325 (9.51%) were identified as homeless people—or having experienced a life episode of homelessness—the database being completed in a longitudinal manner during the patient’s care.

### 2.3. Representativeness and Validation of the Database

First, to validate and investigate the representativeness of our database, we first analyzed whether the different risk factors commonly associated with homelessness in the psychiatric population were also significant risk factors in the REHABase. Thus, differences in the prevalence of homelessness based on known psychiatric risk factors (gender, matrimonial status, employment, criminal history, suicide attempt, substance abuse) were analyzed using contingency tables and successive Pearson’s chi-square tests.

Then, we evaluated the prevalence of homelessness across the different diagnostic categories using the DSM-5 classification. The data were analyzed using successive Pearson’s chi-square tests corrected for multiple comparisons with a Bonferroni correction.

### 2.4. Estimates of Homelessness Predictors’ Importance

#### 2.4.1. Variable Selection

Our main objective was to identify REHABase variables that were associated with homelessness and could be manipulated to identify and reduce the risk of poor housing situations. A subset of 8 easily measurable variables recorded in the REHABase was selected by three clinical investigators (authors MG, CD, and NF): gender, psychiatric diagnosis, pharmacological class of the main treatment, age at onset of symptoms, and results of the GAF, CGI, MARS and BIS rating scales. Gender, which is a classical risk factor for homelessness, was used as the reference predictor for this study in a machine learning model.

#### 2.4.2. Machine Learning Model

To compare the relative importance of several variables in comparison with gender, we used unbiased decision classification and regression trees (unbiased classification and regression trees: CART) [12,23]. The basic goal of a decision tree is to predict the value of a target variable by learning simple decision rules based on different splits of the predictive variables arranged in a flowchart-like structure.

This type of machine learning model is generally easier to understand and interpret than other methods because trees can be visualized; furthermore, the method has many other advantages. It is able to handle both the numerical and categorical data considered and is naturally more robust to outliers than parametric methods, and procedures can be implemented to handle missing data [24].

However, an essential property of classification trees is that the relative predictors’ importance can be estimated. This property can be used for exploratory data analysis, dimensionality reduction, or—as in this study—to identify the most important predictors for establishing surveillance and prevention policies.

#### 2.4.3. Hyperparameter Tuning for Estimates of Predictors’ Importance

The importance of predictors of homelessness was estimated using the ‘fitctree’ and ‘predictorImportance’ functions of the MATLAB R2018a statistics and machine learning toolbox. The ‘interaction-curvature’ option for predictor selection was selected for unbiased estimates of predictor importance. Ref. [25,26] Surrogate splits were used to address missing data and to be insensitive to the order of predictors (option ‘surrogate’, ‘all’). A known problem with machine learning algorithms is that they are likely to stop on trivial and uninformative solutions when the number of elements in the two classes that constitute the training set is strongly unbalanced. For this reason, we chose a 200-fold cross-validation procedure that ensures that the prevalence of homelessness in the training set was always 50%. Then, relative estimates of predictor importance were calculated by dividing all estimates of predictor importance by the median estimate for the gender variable (a statistically significant risk factor for homelessness with low predictive power). Finally, the distributions of the results were presented using a notched boxplot representation (MATLAB R2018a boxplot function, ‘notch’ option) with nonparametric estimates of the differences between medians. To ensure the robustness of our approach with respect to the selection of training parameters and the nature of the investigated data, alternative training was performed and is discussed in the Appendix A.

### 2.5. Post Hoc Analysis of Psychotropic Medications

Among all the variables investigated, psychotropic medication was identified as an important predictive factor of homelessness. Therefore, an additional analysis was carried out to more precisely determine the correlation of psychotropic categories with homelessness in the REHABase (for post hoc analyses of other predictive factors, see Appendix A).

First, our post hoc analysis examined all prescriptions (N = 7260—chi-squared test) and compared two meta-categories: (i) common long-term psychotropic medications (mood stabilizers, antidepressants, antipsychotics) and (ii) medications dedicated to adjuvant therapy (benzodiazepines, nonbenzodiazepine anxiolytics, hypnotics, cyamemazine, and loxapine).

Then, successive chi-squared tests were used to explore the correlations of homelessness with each meta-category, controlling for the false discovery rate and the family-wise error rate (with a Bonferroni correction). We analyzed loxapine separately from the other first-generation antipsychotics and clozapine separately from the other second-generation antipsychotics. Indeed, loxapine has a chemical structure very similar to clozapine, and it exhibits several atypical characteristics. It has a higher affinity toward dopamine D3 than the D2 receptor, and it binds with the D4 receptor with a higher affinity than other dopaminergic receptors, which is similar to clozapine [27]. Moreover, in practice, loxapine is often used by clinicians as a pharmacological treatment of agitation [28,29], whereas clozapine is a unique antipsychotic with an affinity for several receptors indicated for treatment-resistant schizophrenia [30].

### 2.6. Cognitive Factors Exploratory Analysis

In the last part of our analysis, we explored cognitive functioning through different measures collected in the REHABase. Our baseline neuropsychological cognitive assessments included the Wechsler Adult Intelligence Scale—4th edition (WAIS-IV) [31] subscale assessing short-term and working memory (MATRICES), the California Verbal Learning Test (CVLT) [32], RL/RI-16 [33] for global verbal memory, d2-R [34] for selective attention, concentration and speed of processing and the shopping test [35] or Six Element Test [36] for planning abilities. Theory of mind was assessed using the Movie for the Assessment of Social Cognition [37], and attribution style was assessed with the Ambiguous Intentions and Hostility Questionnaire (AIHQ) [38]. For each test, each subscore was Z-transformed along the whole population (N = 3416 users), summed with a weighted sum (+1 of −1 to adjust the direction of the effect), and Z-transformed again. Homoscedasticity was checked with two-sample F tests, and the differences between the mean scores were analyzed using two-sample *t*-tests. The FWER was controlled using Bonferroni’s correction. A confirmatory analysis using nonparametric Wilcoxon rank sum tests, controlling for the FWER, was also implemented.

## 3. Results

### 3.1. Representativeness and Validation of the Database

In the first part of our study, we explored the different individual factors associated with homelessness in the psychiatric population. We replicated common results in the REHABase population: homelessness was significantly associated with male sex, single status, being unemployed, having a criminal history, having a history of suicide attempts, and having a history of substance abuse (mainly cannabis) (Table 2). 

Addictive disorders and schizophrenia spectrum disorders were significantly more strongly associated with homelessness than the other psychiatric disorders (prevalence: 25.86%—Diff 16.34, *p* < 0.001; prevalence: 11.54%—Diff 2.02, *p* < 0.01). On the other hand, neurodevelopmental disorders were significantly less strongly associated with the risk of homelessness than the other diagnostics (prevalence: 3.11%—Diff −6.41, *p* < 0.001). More specifically, the prevalence of homelessness among participants with ASD was 2.12%.

### 3.2. Estimates of Homelessness Predictors Importance

Using a decision tree learning predictive model, we found that all the variables considered had relative predictive values superior to the sex/gender variable (this major risk factor having the lowest predictive power) (see Figure 1A). Among all the variables studied, the most robust variable in terms of predictive value was the nature of the psychotropic medication (sex/gender relative mean predictor importance: 22.8, σ = 3.4). The predictive power of the psychotropic medication was more than 22 times the predictive power of sex/gender and more than two times the predictive power of the diagnosis.

The diagnosis, the age of onset of the first symptoms, and the GAF scale were considered to have the same predictive power (sex/gender relative mean predictor importance: 9.7, σ = 2.6; 9.1, σ = 2.3; 8.7, σ = 2.4, respectively).

The predictive power of the CGI-S scale, the MARS, and the BIS was significantly lower than that of the diagnostic variable (see Figure 1A).

### 3.3. Post Hoc Analysis of the Psychotropic Medication

Then, we investigated psychotropic medication as a correlated factor with homelessness (see Figure 1B). As a first step, we compared the two pharmacological categories: (i) common long-term psychotropic medications (mood stabilizers, antidepressants, antipsychotics) and (ii) medications dedicated to adjuvant therapy (benzodiazepines, nonbenzodiazepine anxiolytics, hypnotics, cyamemazine, and loxapine).

The use of adjuvant treatments (prevalence of homelessness: 12.55%) was significantly more strongly associated with homelessness among our population than the use of long-term psychotropic medication (prevalence of homelessness 10.03%; *p* < 0.001) (Figure 1B). In particular, first-generation antipsychotics (15.61%; *p* < 0.05 FDR corrected), and loxapine (16.57%; *p* < 0.05 FWER corrected) and hypnotics (17.56%; *p* < 0.05 FWER corrected) were significantly more strongly associated with homelessness. On the other hand, antidepressant medication was associated with a protective effect against housing deprivation (9.21%; *p* < 0.05 FWER corrected).

### 3.4. Exploratory Analysis of Cognitive Factors

Among the evaluated cognitive domains, a decrease in the ability of sustainable attention assessed with the d2-R test was significantly associated with homelessness (see Table 3). No other neurocognitive or social cognition variables reached statistical significance.

## 4. Discussion

In the REHABase cohort, homelessness was significantly associated with male sex, single status, being unemployed, having a criminal history, having a history of suicide attempts, and having a history of substance use abuse (mainly cannabis). This first result confirms previous findings among psychiatric populations. These findings suggest that these factors have independent effects on the ability to obtain and maintain decent housing among a psychiatric population.

A recent systematic review and meta-analysis explored the individual predictors of homelessness among the general population [6]. The factors identified in the present study are among the most prominent factors reported in this meta-analysis. Another study [10] of a large psychiatric population identified male gender, presence of substance use disorder, a diagnosis of schizophrenia spectrum disorder or bipolar disorder, and poor functioning as predictors of homelessness. The psychiatric population is also comparable in some aspects to the general population regarding individual risk factors for housing precarity.

We also compared the prevalence of homelessness across different diagnoses. As expected, schizophrenia spectrum disorders and substance abuse were the diagnoses that were most strongly associated with a high prevalence of homelessness, which is consistent with previous studies [10,39,40,41]. On the other hand, the prevalence of neurodevelopmental disorders was significantly lower. Among them, 2.12% of the ASD population was exposed to homelessness, which is still an important but probably underestimated rate. Indeed, the ASD users included in the REHABase are diagnosed adults who are included in a rehabilitation program. However, adults with autism are misdiagnosed, and some of them have poorer quality health care [42].

In the second step of our study, we explored different predictive factors associated with homelessness among the REHABase population (sex, self-stigma and treatment compliance, GAF score, diagnosis, age of first symptoms, severity of symptoms, and psychotropic medication). Using classification trees and criteria for assessing the importance of predictors, we confirm that all these factors have a relative importance greater than the sex/gender variables, a well-established risk factor for homelessness but with an assumed weak predictive power. Nevertheless, among all the variables analyzed, psychotropic medication was largely designated as a predictor of major importance. When comparing the different pharmacological classes, loxapine and hypnotics were both significantly associated with a higher risk of homelessness. Indeed, hypnotics are a well-known pharmacological class exposed to a risk of tolerance and dependence, manifested through the use of high doses over long periods [43]. Hypnotic agents are often prescribed to improve sleep in the psychiatric population. According to our data, these prescriptions are potentially risky for homelessness and should therefore be monitored very closely. Loxapine is a conventional antipsychotic that is considered an atypical antipsychotic at low doses but has an important sedative effect at higher doses. Therefore, it is commonly used as an acute treatment of agitation associated with schizophrenia or other psychiatric disorders [27]. Here, again, the question of polyprescription must be considered cautiously in the psychiatric population—in particular, limiting the long-term effects of sedatives, which lead to desocialization and homelessness. It can be argued that long-term sedation diminishes a person’s ability to overcome social adversity. Frequently, polymedications combine benzodiazepines with antipsychotics in cases of anxiety symptoms, hypnotics for insomnia, and several antipsychotics for antiaggresion effects, especially in cases of dangerousness [44,45]. The prevalence of antipsychotic polypharmacy varies, and the incidence is higher in Europe and Asia than in North America [46]. Antipsychotic polymedications have been associated with an increased burden of side effects, worsened adherence due to treatment complexity, higher total dosages, increased risk of drug–drug interaction, and medication errors [47,48,49,50]. We can therefore hypothesize that a sedative polyprescription (especially with first-generation antipsychotics and hypnotics) weakens the person’s capacity for empowerment and social adaptation.

Interestingly, exploratory analysis of cognitive factors also appears to independently validate this hypothesis. Among all the evaluations, the d2-R test is particularly associated with homelessness by assessing increased attentional difficulties in this population. In particular, difficulties in sustained attention may be associated with the sedative effects of certain treatments.

On the other hand, we found a potential protective impact of antidepressant and mood stabilizer prescriptions on housing deprivation in psychiatric populations. This could lead to proposing a systematic screening and treatment of mood disorders in the homeless population. A previous study using a machine learning model reported that self-reported lifetime histories of depression, the trauma of having a loved one murdered, and posttraumatic stress disorder were the three strongest predictors of homelessness in U.S. army soldiers [51], as observed in a recent meta-analysis in homeless people [8]. Indeed, screening for depression could be the first step toward social reintegration. Moreover, notably because of addictive and psychotic comorbidities, mood disorders, particularly depression, could be largely underestimated among the homeless population.

In the last part of our study, we explored the neurocognitive and social cognitive factors associated with homelessness. Attention deficit was the only variable significantly associated with homelessness in our study and, as mentioned earlier, might be partially associated with the sedative effects of some treatments. Previous research suggests that cognitive impairment is overrepresented in the homeless population [13]. A recent study found that cognitive impairment was evident in 80% of a precariously housed young population aged between 16 and 24 years [52]. As observed in our study, the most frequent difficulties involved attention and processing speed. Moreover, the few studies that have explored specific cognitive fields in homelessness found that attention was consistently impaired among this population. These results suggest that cognitive factors should be closely monitored among the psychiatric population, especially in the case of homelessness. Further studies are needed to determine whether appropriate cognitive remediation therapy could be used to reduce the risk of homelessness [53].

Finally, our study has some limitations. First, it is an exploratory study, and thus, it cannot establish causal links. However, it does provide interesting avenues to investigate predictors of homelessness among the psychiatric population. Second, our study included a specific population of psychiatric users who were referred to rehabilitation centers, leading to selection bias. Nevertheless, our results enable us to compare users across various diagnostic categories to identify, through multivariate analysis, the most relevant predictive factors, as the inclusion criteria were similar across diagnostic classes. Third, the assessment of homelessness is based on medical interviews, which can vary from one patient to another. Indeed, some users may be reluctant to reveal that they are homeless.

## 5. Conclusions

With some limitations, the current REHABase study used decision classification and regression trees to effectively identify the most relevant predictors of homelessness, which can be useful for the implementation of optimized and effective prevention policies. Specifically, this analysis allowed us to identify that the sedative or antidepressant properties of pharmacological interventions had a much greater influence on patients’ autonomy and quality of life than assumed. The increase in studies and knowledge makes it increasingly easy to identify the correlates of a societal, environmental, or public health problem. However, implementing effective strategies to address these problems remains complicated. Mining large databases with appropriate tools to identify and select critical variables can be effective in better understanding an increasingly data-rich world, where it is easy to get lost in strategies that ultimately may not focus on the most critical and effective factors.

## Figures and Tables

**Figure 1 ijerph-19-12268-f001:**
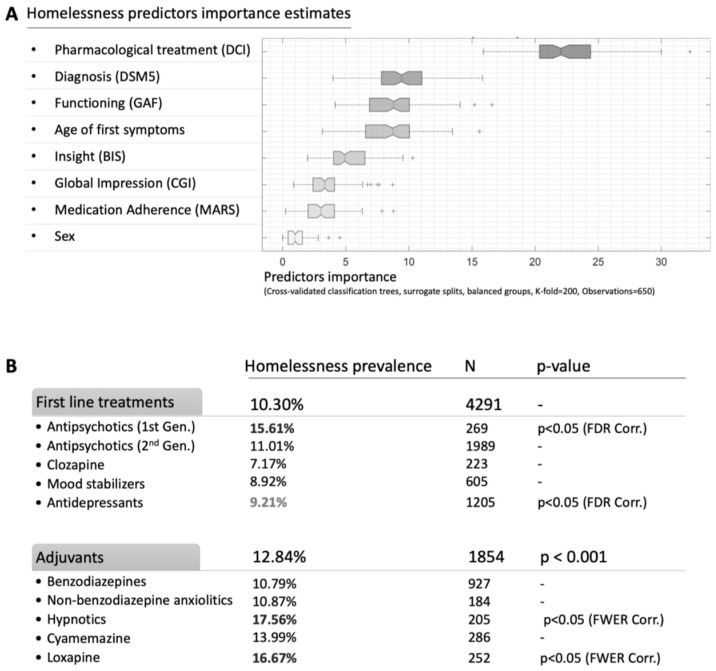
(**A**,**B**): Estimates of the importance of predictors of homelessness.

**Table 1 ijerph-19-12268-t001:** Main psychiatric diagnoses.

*Psychiatric Diagnoses (DSM-5 Criteria)*	*N*	*%*
*Schizophrenia spectrum disorders*	1598	45.70%
*Neurodevelopmental disorders*	443	12.67%
*Bipolar disorders*	407	11.64%
*Personality disorders*	372	10.64%
*Depressive disorders*	230	6.58%
*Anxiety disorders*	200	5.72%
*Addiction disorders*	58	1.66%
*Post-traumatic stress disorders*	54	1.54%
*Other diagnostic categories*	135	3.86%

**Table 2 ijerph-19-12268-t002:** Analysis of usual risk factors of homelessness in REHABase cohort.

	*Factor*	*Homelessnes Prevalence (%)*	*Mean Prevalence (%)*	*Difference with Mean Prevalence (%)*	*p (chi2)*
**Gender**	Female	**7.41**	9.51	**−2.1**	***p* < 0.01**
	Male	**10.71**	9.51	**1.2**	***p* < 0.01**
**Matrimonial status**	Single	**10.16**	9.49	**0.67**	***p* < 0.05**
	Divorced	10.73	9.49	1.24	
	Married	**4.18**	9.49	**−5.31**	***p* < 0.01**
	Common-law marriage	7.48	9.49	−2.01	
**Employment**	Job	**4.40**	9.77	**−5.37**	***p* < 0.001**
	No employ	**10.35**	9.77	**0.58**	***p* < 0.001**
**Criminal History**	Without	**7.55 **	9.54	**−1.99 **	***p* < 0.001 **
	With	** 24.38 **	9.54	**14.84 **	***p* < 0.001 **
**Suicidal attempt history**	No	**8.33**	9.58	**−1.25**	***p* < 0.001**
	Yes	**12.59**	9.58	**3.01**	***p* < 0.001**
**Substance abuse**	No	**5.64**	9.58	**−3.94**	***p* < 0.001**
	Tabaco	9.15	9.58	−0.43	
	Tabaco, alcohol	11.73	9.58	2.15	
	Tabaco, alcohol, cannabis	**21.65**	9.58	**12.07**	***p* < 0.001**
	Tabaco, cannabis	**18.92**	9.58	**9.34**	***p* < 0.001**
	Alcohol	7.44	9.58	−2.14	
	Others	**15.74**	9.58	**6.16**	***p* < 0.001**

**Table 3 ijerph-19-12268-t003:** Neurocognitive and social cognitive evaluations (Scores Homeless—Scores Non-Homeless).

TEST (Z scores)	MATRICES	SIMILITUDES	RLRI16	CVLT	MEM	d-2R	ACSo	MASC	AIHQ
** Diff. Mean Scores **	0.07	0.14	0.11	0.24	0.04	0.30	−0.02	0.17	−0.14
** Diff. Median Scores **	0	0.15	0.14	0.40	−0.01	0.25	0	0.16	−0.19
** Pval ** ** (T test) **	0.56	0.27	0.19	0.12	0.62	0.0011	0.85	0.15	0.38
** FWER corr. **	NS	NS	NS	NS	NS	*p* < 0.05	NS	NS	NS
** Pval (Wilcoxon) **	0.45	0.23	0.06	0.11	0.55	0.0017	0.73	0.14	0.41
** FWER CORR. **	NS	NS	NS	NS	NS	*p* < 0.05	NS	NS	NS

## Data Availability

All data are available upon request to Nicolas Franck (nicolas.franck@ch-le-vinatier.fr).

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
