# Peer review of "Actionable Predictive Factors of Homelessness in a Psychiatric Population: Results from the REHABase Cohort Using a Machine Learning Approach"

_ijerph, 2022, doi:10.3390/ijerph191912268_

Round 1
Reviewer 1 Report
The article is quite interesting and the subject matter is explained in a clear and complete way. The analysis of the predictive factors is complete and the efforts of the authors are completely effective. However, I recommend changing the order of the estimates of the importance of predictors of homelessness on page 8 as follows: Diagnosis; Pharmacological treatment; Functioning; Age of first symptoms; Insight; Global impression.
Author Response
We thank the reviewer for such positive feedback. Regarding the order of predictors presented in the figure of results, it is indeed possible to present it as suggested by the reviewer with diagnosis first - diagnosis being generally the factor considered the most predictive apriori and directing public health policies. The alternative is to present the results in order of measured importance as presented in this version of the paper. As our first approach is strictly data-driven, we prefer to keep the presentation of the results in order of measured importance in order to avoid possible confusion between results and apriori hypotheses.

Reviewer 2 Report
The submitted review discussed about the predictive factors of homelessness in a psychiatric population using data from the REHABase, which is a cohort of patients with psychiatric or neurodevelopmental disorders referred to French psychosocial rehabilitation centres.
The authors summarized data related to different risk factors previously associated with homelessness in mental health, including (sex, self-stigma and treatment compliance, GAF score, diagnosis, age of first symptoms, severity of symptoms and psychotropic medication
Although the manuscript presents an extensive literature overview, some points must be carefully considered.
1. Author may need to recheck the term of "non-heterosexual sexual orientation" on page 2, line 65.
2. Was the cohort being diagnosed with psychiatric disorders before being homeless or vice versa? Will these two different conditions influence the conclusion?
3. What kinds of algorithms in machine learning approach that apply in this study? How many algorithms the authors have tried for this study?
4. The conclusion section should be extended. Are there any suggestions and further directions related to the quality of studies on this topic?
Author Response
The submitted review discussed about the predictive factors of homelessness in a psychiatric population using data from the REHABase, which is a cohort of patients with psychiatric or neurodevelopmental disorders referred to French psychosocial rehabilitation centres.
The authors summarized data related to different risk factors previously associated with homelessness in mental health, including (sex, self-stigma and treatment compliance, GAF score, diagnosis, age of first symptoms, severity of symptoms and psychotropic medication
Although the manuscript presents an extensive literature overview, some points must be carefully considered.
- Author may need to recheck the term of "non-heterosexual sexual orientation" on page 2, line 65.
Thank you for raising this point. Indeed, Nilsson et al. use the term “non-heterosexual identity” and not “non-heterosexual sexual orientation”. The correction was made in the manuscript.
- Was the cohort being diagnosed with psychiatric disorders before being homeless or vice versa? Will these two different conditions influence the conclusion?
This is indeed an important point. If we know the date of diagnosis for each patient, the temporality of the episode of precariousness is not indicated in the database. But for most of the patients (neurodevelopmental disorders and schizophrenia essentially – see supplementary analysis 3) the diagnosis is relatively early and generally precedes the episode of homelessness.
For this reason, the variable 'age of first symptoms' was considered in the main analysis and was more specifically analyzed in the supplementary analysis 3. The result is that this factor is poorly predictive of homelessness (main results) and we could not find a significant effect of this factor on the prevalence of homelessness (Supplementary Analysis 3). This condition therefore probably has a limited influence on the conclusion.
The paragraph 'Homelessness in the Rehabase Database' has nevertheless been slightly modified to clarify that the temporality of homelessness was not strictly taken into account in the study.
- What kinds of algorithms in machine learning approach that apply in this study? How many algorithms the authors have tried for this study?
Since the data are heterogeneous (numerical and categorical) and the amount of missing data can vary between the variables analyzed, only regression/classification tree type algorithms are optimized for this type of analysis. The choice of the analysis algorithm was therefore straightforward – this point has been presented in the paragraph "machine learning model".
Therefore, only one algorithm was used in this study. But multiple parameterizations of the algorithm were considered in order to check the robustness of the approach and the results (see supplementary materials). Finally, only the most robust and systematically reproduced results were presented and discussed.
- The conclusion section should be extended. Are there any suggestions and further directions related to the quality of studies on this topic?
Thanks for the suggestion, we have added the following paragraph at the end of the discussion:
“Specifically, this analysis allowed us to identify that the sedative or antidepressant properties of pharmacological interventions had a much greater influence on patients' autonomy and quality of life than assumed. The increase in studies and knowledge makes it increasingly easy to identify the correlates of a societal, environmental or public health problem. But implementing effective strategies to address these problems remains complicated. Mining large databases with appropriate tools to identify and select critical variables can be effective in better understanding an increasingly data-rich world, where it is easy to get lost in strategies that ultimately may not focus on the most critical and effective factors.”

Reviewer 3 Report
This manuscript investigates predictive factors of homelessness in a psychiatric population using the REHABase cohort. As for exploratory analysis, the authors utilize a machine learning approach to better classify meaningful and actionable factors, which have a potential for paying effective attention to reduce likelihood to become homelessness.
The contents are good enough to provide information in the database demographics and the analysis method is well described. The main results are shown numerically and in a visually guided manner. The main topics are well discussed in paragraph writing; the hypothesis and the conclusion are consistent.
Therefore, I think the manuscript is acceptable except some major concerns and minor points as shown below,
Major concerns
Introduction
The latter part seems to be written in a mixed form of the methods and the methodological discussions, not the introduction why the authors utilized the REHABase nor why the machine learning exploration was selected. I believe that the authors should describe the hypothesis more clearly by using the topic sentences to make the methods rational for interested readers.
Materials and methods
The descriptions why the CART machine learning model utilized in the present study should be moved to the introduction.
I think that the interested readers would like to know the computing environment, main PCs specifications, and total calculation time for the present analysis.
Results
Table 2 should show both the homeless prevalence and the mean prevalence.
Figures
Figure A lacks information on how these error bars can be interpreted (e.g., SD or SE, or others) and what the predictors importance indicates for lay readers
What is the reason for a difference of statistical significance utilized; FDR corrections in First line treatments and FWER corrections in adjuvants?
Discussion
More specific discussion is warranted about the significant differences only in d2-R scores.
Minor points
1) There are large amounts of mixture of different fonts in all the manuscript.
2) A full expression of the REHABase needs to be shown at the first presentation of this word.
I hope you may kindly consider my suggestion. Thank you.
Author Response
his manuscript investigates predictive factors of homelessness in a psychiatric population using the REHABase cohort. As for exploratory analysis, the authors utilize a machine learning approach to better classify meaningful and actionable factors, which have a potential for paying effective attention to reduce likelihood to become homelessness.
The contents are good enough to provide information in the database demographics and the analysis method is well described. The main results are shown numerically and in a visually guided manner. The main topics are well discussed in paragraph writing; the hypothesis and the conclusion are consistent.
Therefore, I think the manuscript is acceptable except some major concerns and minor points as shown below,
Major concerns
Introduction
The latter part seems to be written in a mixed form of the methods and the methodological discussions, not the introduction why the authors utilized the REHABase nor why the machine learning exploration was selected. I believe that the authors should describe the hypothesis more clearly by using the topic sentences to make the methods rational for interested readers.
Materials and methods
The descriptions why the CART machine learning model utilized in the present study should be moved to the introduction.
Thanks for these suggestions, in order to make the analysis strategy of the article clearer, a paragraph introducing it has been added to the introduction:
“Our strategy for identifying these critical actionable factors was as follows. First, to identify actionable factors that were easily identified in the target population. Then, to identify multi-diagnostic databases listing these factors. Finally, the core of this study was to design an analysis strategy based on machine learning and robust to the heterogeneity of the data in order to comparatively evaluate the different factors identified. The final objective was to rank the importance of the different factors identified in order to analyze in detail the essential factors to be taken into account when designing the most effective public health strategies. Indeed, this study aimed to identify the most critical predictors of homelessness to provide guidance for further research and tailored prevention and intervention programs related to mental health. “
and the latter has been reworked to better introduce the CART method.
I think that the interested readers would like to know the computing environment, main PCs specifications, and total calculation time for the present analysis.
Unlike methods such as deep-learning which may require complex parallel infrastructures and long computation times in the learning phases, the complexity of regression tree/classification methods is low. The computation time is of the order of the second for standard machines. This is why this question has not been addressed. But in the introduction, we added a reference to the calculation speed of the method.
Results
Table 2 should show both the homeless prevalence and the mean prevalence.
The table has been modified to have a direct reading of the mean prevalence measured for each category analyzed.
Figures
Figure A lacks information on how these error bars can be interpreted (e.g., SD or SE, or others) and what the predictors importance indicates for lay readers
There are no error bars in the result figure - part A. But the whole distribution of values has been represented with box-plot representations which show the median, the first and the third quartiles as well as the extrema of the distributions and the outliers:
“boxplot draws points as outliers if they are greater than q3 + w × (q3 – q1) or less than q1 – w × (q3 – q1), where w is the multiplier Whisker, and q1 and q3 are the 25th and 75th percentiles of the sample data, respectively.
The default value for 'Whisker' corresponds to approximately +/–2.7σ and 99.3 percent coverage if the data are normally distributed. The plotted whisker extends to the adjacent value, which is the most extreme data value that is not an outlier.”
Ref : https://fr.mathworks.com/help/stats/boxplot.html
The text in the method section has been slightly modified to clarify this point :
“Finally, the distributions of the results were presented using a notched boxplot representation (MATLAB R2018a boxplot function, ‘notch’ option) with nonparametric estimates of the differences between medians.”
What is the reason for a difference of statistical significance utilized; FDR corrections in First line treatments and FWER corrections in adjuvants?
Both FDR and FWER were used in the post hoc analysis of psychotropic medications. The FDR correction was used so as not to have too large type 1 error correction that would lead to too large type 2 error probability that could mask possible positive pharmacological effects on homelessness. This is particularly important in this analysis where we are trying to explain the different contributions of the different molecules on the observed magnitude of predictor importance. Contributions than can be beneficial or deleterious.
Discussion
More specific discussion is warranted about the significant differences only in d2-R scores.
We agree with the reviewer that there was little discussion of the specific effect on the d2 test, whereas the associations of d2/homelessness and pharmacology/homelessness might be due to the issue of the sedative effect of certain treatments. The two assessments being independent in the design of the analysis it is not possible to assess a direct statistical association but it is clearly something to discuss. A paragraph referring to this possible association has been added to the discussion.
Minor points
- There are large amounts of mixture of different fonts in all the manuscript.
Thank you, this has been checked and corrected in the submitted document.
- A full expression of the REHABase needs to be shown at the first presentation of this word.
REHABase is not really an acronym, more a combination of the two words Rehabilitation and Database. The first mention of this word in the abstract has been modified to clarify this point.
I hope you may kindly consider my suggestion. Thank you.
Thank you for your suggestions, hoping to have answered your requests.
